# Development of Hydrolysis and Defatting Processes for Production of Lowered Fishy Odor Hydrolyzed Collagen from Fatty Skin of Sockeye Salmon (*Oncorhynchus nerka*)

**DOI:** 10.3390/foods10102257

**Published:** 2021-09-23

**Authors:** Krisana Nilsuwan, Kasidate Chantakun, Lalita Chotphruethipong, Soottawat Benjakul

**Affiliations:** International Center of Excellence in Seafood Science and Innovation, Faculty of Agro-Industry, Prince of Songkla University, Hat Yai, Songkhla 90110, Thailand; krisana.n@psu.ac.th (K.N.); kasidate.c@psu.ac.th (K.C.); lalita.ch@psu.ac.th (L.C.)

**Keywords:** salmon skin, hydrolyzed collagen, fat removal, fishy odor

## Abstract

Lipid oxidation has a negative impact on application and stability of hydrolyzed collagen (HC) powder from fatty fish skin. This study aimed to produce fat-free HC powder from salmon skin via optimization of one-step hydrolysis using mixed proteases (papain and Alcalase) at different levels. Fat removal processes using disk stack centrifugal separator (DSCS) for various cycles and subsequent defatting of HC powder using isopropanol for different cycles were also investigated. One-step hydrolysis by mixed proteases (3% papain and 4% Alcalase) at pH 8 and 60 °C for 240 min provided HC with highest degree of hydrolysis. HC powder having fat removal with DSCS for 9 cycles showed the decreased fat content. HC powder subsequently defatted with isopropanol for 2 cycles (HC-C9/ISP2) had no fat content with lowest fishy odor intensity, peroxide value, and thiobarbituric acid reactive substances than those without defatting and with 1-cycle defatting. HC-C9/ISP2 had high *L**-value (84.52) and high protein (94.72%). It contained peptides having molecular weight less than 3 kDa. Glycine and imino acids were dominant amino acid. HC-C9/ISP2 had Na, Ca, P, and lowered odorous constituents. Combined processes including hydrolysis and defatting could therefore render HC powder free of fat and negligible fishy odor.

## 1. Introduction

Hydrolyzed collagen (HC), especially from fish skin or scale, has drawn attention as a promising source of biologically active peptides [1]. HC has been documented to induce skin nourishment [2] and bone strengthening [3]. HC from fish skins is acceptable as Kosher and Halal product [4]. Skins of fish including unicorn leatherjacket (*Aluterus monoceros*) [5], Asian seabass (*Lates calcarifer*) [6], etc. have been used for production of hydrolyzed collagen. In general, an enzymatic process is commonly used due to the mild condition with less side effects [7]. The optimization of pretreatment or enzymatic hydrolysis has been carried out to obtain the high yield, along with the desired characteristics or quality [8]. Under the controlled enzymatic hydrolysis, the type of enzyme as well as hydrolysis process had the influence on the degree of hydrolysis of the resulting hydrolysate [9,10]. Benjakul, Karnjanapratum, and Visessanguan [1] reported that HC prepared from seabass (*Lates calcarifer*) skin using 3% (*w*/*w*, based on solid content) papain in the first step of hydrolysis, followed by using 2% (*w*/*w*, based on solid content) Alcalase for the second step, consisted of antioxidative peptides. However, little information on the use of mixed proteases within one step for HC production exists.

Salmon is globally one of the most popular fish species for consumption due to its high nutritive value and high bioactive pigment, namely astaxanthin. Salmon meat contains high quality protein with a high amount of essential amino acids [11]. Additionally, salmon meat has an attractive taste, smell, and color. The salmon processing industry mostly disposes of a large amount of by-products, such as skins (about 7%) generated during deskinning [11]. Enzymatic hydrolysis has been widely used to produce protein hydrolysate with high protein recovery from salmon skin including farmed salmon, Atlantic salmon, and salmon (*Salmo salar*) [12,13,14]. Via the optimal hydrolytic process, the resulting hydrolysate contained the peptides with high bioactivity such as antioxidant activity, proliferation of fibroblast, and osteoblast cells [1,2,3]. However, salmon skin has a high fat content (23.3–61.5% dry basis) and is rich in polyunsaturated fatty acids (PUFA) [11,15]. The oxidation of PUFA associated with skin matrix is a drawback associated with undesirable characteristic, particularly fishy odor/flavor and instability of HC. Several techniques have been developed to reduce fat content of fish skin. Sae-leaw, et al. [16] documented that citric acid pretreatment, followed by defatting with 30% isopropanol, could remove fats and fishy odor in seabass skin gelatin. Chotphruethipong, et al. [17] reported that pulsed electric field (PEF) in conjunction with pancreas lipase (25 U/g dry matter) with the aid of vacuum impregnation could remove 86.93% fats from seabass skins. As a consequence, HC prepared from defatted skin showed much lower fishy odor and flavor [18]. Although some fat can be removed from skin, the fat retained is released and present in HC, especially in form of emulsion, leading to an undesirable fishy odor.

To reduce the use of solvent and time consumed, the use of disk stack centrifugal separator (DSCS) could be employed. Jukkola, et al. [19] documented that DSCS could separate fat globules from milk up to 93% when the centrifugal number was increased to 3 cycles. Nonetheless, no information regarding the use of DSCS for fat removal of HC from fish skin, especially that produced using mixed proteases for one-step hydrolysis process, has been reported. Furthermore, to increase oxidative stability of HC, appropriate solvent could be further used to extract the fat residue in HC powder. Therefore, this study aimed to develop the hydrolysis of salmon skin using mixed proteases for one-step process, and to investigate fat removal processes using DSCS and subsequent solvent extraction to yield HC powder lacking in fat and fishy odor.

## 2. Materials and Methods

### 2.1. Enzymes and Chemicals

Alcalase (EC 3.4.21.62) (2.4 unit/g) from *Bacillus licheniformis* and papain from papaya (*Carica papaya*) latex (E.C.3.4.22.2) (3 units/g) were obtained from Siam Victory Chemicals Co, Ltd. (Bangkok, Thailand). 2,4,6-trinitrobenzenesulphonic acid (TNBS) was procured from Sigma-Aldrich Chemical Co. (St. Louis, MO, USA). All chemicals were of analytical grade.

### 2.2. Preparation of Salmon Skin

Skins of sockeye salmon (*Oncorhynchus nerka*) collected from the Nissui (Thailand) Co., Ltd., Songkhla, Thailand were stored in ice (1:2, *w*/*w*), in which a polystyrene box was used as a container. Within 1 h of transportation, the skins reached the International Center of Excellence in Seafood Science and Innovation. The remaining meat was removed from the skin manually and the skin was washed with cold water (≤10 °C). The prepared skins were stored at −20 °C under the vacuum in polyethylene bags (less than 1 month).

Before use, the size of frozen skins was reduced to small pieces (3.0 × 3.0 cm^2^) using an electric sawing machine. The prepared samples (3.0 × 3.0 cm^2^) were then subjected to non-collagenous protein removal process as tailored by Benjakul, Karnjanapratum and Visessanguan [1]. Skins were mixed with 0.05 M NaOH (1:10, *w*/*v*) and stirred gently at 150 rpm. After 1 h, the soaking solution was drained, and the alkali pretreatment was performed in the same manner for a total of three times. Alkali-treated skins were washed with water until neutralized. Thereafter, the prepared skin was soaked in 0.05 M citric acid (1:10, *w*/*v*) with gentle agitation for 15 min and left for 45 min. The swollen skins were rinsed with water until a neutral pH was reached. All the operations were run at room temperature (RT; 28–30 °C).

### 2.3. Effect of Hydrolysis Conditions on Degree of Hydrolysis of HC

#### 2.3.1. Effect of Mixed Proteases at Various Concentrations and Different pHs

The swollen skins were mixed with distilled water (1:5, *w*/*v*) and adjusted to pH 7 or 8 using 1.0 M HCl or 1.0 M NaOH. Mixed papain (P) and Alcalase (A) at varying concentrations (*w*/*w*, based on solid content) included 3%P + 3%A, 3%P + 4%A, 4%P + 3%A, and 4%P + 4%A. The aforementioned mixed proteases were added in prepared mixture and incubated in a temperature-controlled water bath at 50 °C with continuous stirring using a propeller. During hydrolysis up to 270 min, the samples (1 mL) were taken and added with hot 5% SDS (85 °C) at a 1:1 (*v*/*v*) ratio and heated at 85 °C for 30 min to terminate reaction and solubilize total proteins and peptides. Degree of hydrolysis (DH) was measured as tailored by Benjakul and Morrissey [20]. Hydrolysis condition (mixed proteases, hydrolysis pH and time) yielding the highest DH was chosen for further study.

#### 2.3.2. Effect of Temperature

Swollen skins mixed with distilled water were prepared as previously detailed and pH was adjusted to 8. The selected mixed proteases as described above (3%P + 4%A) were added into the mixture. Hydrolysis was run at 50, 55, and 60 °C up to 270 min. The samples were taken and added with hot SDS solution, as previously described. Hydrolysis temperature rendering the highest DH was chosen. 

The selected HC was then lyophilized using a freeze-dryer (Model Duratop™ lP/Dura Dry™ lP, FTS^®^ System, Inc., Stone Ridge, NY, USA) at −50 °C for 72 h. To determine fatty acid profile of resulting HC powder, lipid was extracted from the powder using Bligh and Dyer [21] method. Transmethylation was done with the aid of 2 M methanolic NaOH and 2 M methanolic HCl to attain fatty acid methyl esters (FAMEs) as described by Muhammed, et al. [22]. The Agilent 7890B (Santa Clara, CA, USA) gas chromatography system connected with flame ionization detector (FID) was used and the condition for separation was used as detailed by Nilsuwan, et al. [23].

### 2.4. Effect of Centrifugal Cycles on Fat Removal of HC

HC was prepared from swollen skins using 3%P + 4%A at pH 8 for 240 min, followed by heating at 90 °C for 15 min. HC solution was cooled to RT and subjected to fat removal using the disk stack centrifugal separator (SPX FLOW Technology Italia S.p.A., Milan, Italy) with the feed rate of 2.0 L/min for 0, 3, 6 and 9 cycles. The pressure of heavy phase (hydrolysate phase) and light phase (fat phase) was 2.0 and 0.5 bar, respectively. HC solutions subjected to fat removal for different centrifugal cycles were collected and lyophilized. Lyophilized HC samples were analyzed. 

#### 2.4.1. Fat Content

HC powder without and with fat removal for different centrifugal cycles were determined for fat content as per the method of AOAC [24] using Soxhlet apparatus (Soxterm–Gerhardt, Bonn, Germany). Petroleum ether was used as a solvent. 

#### 2.4.2. Color

Color parameters consisting of *L** (Lightness/darkness), *a** (redness/greenness) and *b** (yellowness/blueness) were determined using a Hunter Lab Colorimeter [1]. ∆*E** (color difference) was also examined.

### 2.5. Effect of Isopropanol on Fat Removal of HC Powder

HC defatted using DSCS for 9 cycles showing the lowest fat content and highest *L** value was selected and lyophilized, so called ‘HC powder’. HC powder was mixed with isopropanol at a solid/solvent ratio of 1:10 (*w*/*v*). The mixture was further mixed with the aid of ultrasonic processor using a 13-mm probe at an amplitude of 70% with pulse mode (on/off at 10 s) for 10 min. The mixture was then stirred at 150 rpm for 20 min with a magnetic stirrer at 4–8 °C and subsequently centrifuged at 10,000× *g* for 20 min at 4 °C. HC was collected and the defatting process was conducted in the same manner for 2 and 3 cycles. All the collected HC samples defatted with isopropanol for 1, 2, and 3 cycles were spread on the tray for evaporation at 25 °C for 30 min with the aid of blowing air. All HC powders were subjected to analyses.

#### 2.5.1. Fat Content and Color

Fat content [24] and color [1] were determined as mentioned in Section 2.4.1 and Section 2.4.2, respectively. 

#### 2.5.2. Fishy Odor Intensity (FOI)

FOI was determined following the procedure of Sae-leaw, Benjakul, and O’Brien [16] using 10 trained panelists. Panelists were trained with standards as detailed by Sae-leaw, Benjakul, and O’Brien [16]. Ten-day iced-stored salmon skin was hydrolyzed under the selected condition and the resulting HC was used as a standard of fishy odor. The standard HC was mixed with water to obtain concentration of 0, 0.5, and 1% (*w*/*v*) representing the score of 0 (none), 2 (medium), and 4 (strong), respectively. HC powder sample was dissolved in water to have a final concentration of 0.75%. Solution was placed in sealable cup. The panelists were asked to sniff the headspace above the samples and assessed for FOI.

#### 2.5.3. Peroxide Value (PV) and Thiobarbituric Acid Reactive Substances (TBARS)

The method of Richards and Hultin [25] was adopted for PV measurement. Cumene hydroperoxide (0–40 ppm) was used as standards and PV was reported as mg cumene hydroperoxide equivalents/100 g dry HC. TBARS values were also determined [26]. 1,1,3,3-tetramethoxypropane (MDA) (0–10 ppm) was used as standards. TBARS value was expressed as mg MDA equivalents/100 g dry HC.

### 2.6. Characterization of Selected Defatted HC

HC with the lowest fat content and FOI (HC powder defatted with isopropanol for 2 cycles) was characterized.

#### 2.6.1. Size Distribution

Molecular weight of HC powder was determined by MALDI-TOF following the method of Benjakul, Karnjanapratum and Visessanguan [2]. The Autoflex Speed MALDI-TOF (Bruker, GmbH, Bremen, Germany) mass spectrometer equipped with a 337 nm nitrogen laser was used.

#### 2.6.2. Amino Acid Composition

HC powder was treated with 4.0 M methanesulphonic acid at 110 °C for 22 h under the reduced pressure to prevent the oxidation of tryptophan. The sample was neutralized with 3.5 M NaOH and diluted with 0.2 M citrate buffer (pH 2.2). An aliquot (100 µL) was injected into an amino acid analyzer (JLC-500/V AminoTac™, JEOL USA Inc., Peabody, MA, USA). 

#### 2.6.3. Proximate Composition and Mineral Content

Protein and ash contents of HC powder defatted with isopropanol for 2 cycles was analyzed using AOAC analytical methods no. 920.153 and 928.08, respectively [24]. Inductively coupled plasma optical emission spectrometer (ICP-OES) (Model AVIO 500, Perkin Elmer, Shelton, CT, USA) was used for measurement of Na, Ca, and P contents in HC as detailed by Benjakul, et al. [27].

#### 2.6.4. Volatile Compounds

HC powder defatted with isopropanol for 2 cycles was analyzed for volatile compounds using a solid-phase microextraction gas chromatography mass spectrometry (SPME GC-MS) [28] in comparison with the HC defatted using centrifugal separator for 9 cycles and HC without fat removal. The abundance of individual compound was recorded.

### 2.7. Statistical Analysis

Completely randomized design (CRD) was used for the entire study. Experiment and analysis were conducted in triplicate (*n* = 3). Analysis of variance (ANOVA) was done and the differences among the samples were determined using Duncan’s multiple range test at the *p* < 0.05 level. The analysis was performed with a SPSS package (SPSS for windows, SPSS Inc., Chicago, IL, USA).

## 3. Results and Discussion

### 3.1. Effect of Hydrolysis Conditions on Degree of Hydrolysis and Fatty Acid Composition in HC

#### 3.1.1. Degree of Hydrolysis (DH)

DHs of HC from swollen salmon skin, as affected by mixed proteases at various concentrations as a function of hydrolysis time at pH 7 and 8, are shown in Figure 1. Generally, the DH of all HC samples increased with augmenting hydrolysis time. High hydrolysis rate was noted within the first 30 min, reflecting that numerous peptide bonds were cleaved. Lower rate of hydrolysis was subsequently obtained and reached a plateau. Those changes were depending on proteases and pH used. At pH 7, the hydrolysis times to reach the plateau were 120, 150, 90 and 90 min when 3%P + 3%A, 3%P + 4%A, 4%P + 3%A and 4%P + 4%A were applied, respectively. When pH 8 was used, hydrolysis reached the plateau at 180, 240, 150, and 150 min as 3%P + 3%A, 3%P + 4%A, 4%P + 3%A, and 4%P + 4%A were applied, respectively. This result suggested that the decreased hydrolysis rate was more likely due to a depletion of substrate, enzyme autodigestion, and/or product inhibition [7]. Overall, hydrolysis pattern was governed by types of proteases and pH used.

When comparing DH of HC prepared at pH 7 and 8, higher DH was observed for that prepared at pH 8 (*p* < 0.05), regardless of the concentration of mixed proteases and hydrolysis time used. This result suggested that the mixed proteases were more active at alkaline pH. Commonly, papain was capable of hydrolyzing peptide bonds at optimum pH of 7–7.5, while Alcalase showed a broad activity in alkaline pH range (8–10) [10]. Adler-Nissen [29] documented that pH had the influence on the substrate and enzyme by changing the charge distribution and conformation of the molecules. As a result, pH was a vital factor for determining DH.

Moreover, the highest DH was observed for HC prepared at pH 8 with the aid of 3%P + 4%A for 240 min. Under the optimum hydrolysis condition, mixed proteases (both papain and Alcalase) might hydrolyze swollen skin with loosened structure. As a consequence, a number of peptides were released and present in the hydrolysate. Chotphruethipong, et al. [30] documented that papain was commonly able to hydrolyze peptide bonds that involve glycine, basic amino acids or leucine, while Alcalase prefers to cleave peptide bonds containing hydrophobic residues [31]. In the presence of both proteases, which had broad specificity, peptide bonds were cleaved effectively as witnessed by the marked increase in DH.

Figure 2 shows the DH of HC from salmon skin hydrolyzed at pH 8 with the selected mixed proteases (3%P + 4%A) at different temperatures (50, 55, and 60 °C). Among all temperatures used, the HC hydrolyzed at 60 °C showed the highest DH (*p* < 0.05) when hydrolysis time of 240 min was used. This result indicated that the optimal hydrolysis temperature could favor hydrolysis, thus producing the different peptides in HC. Klomklao and Benjakul [10] found that temperature had a profound impact on the activity of enzymes. In addition, temperature higher than denaturation temperature could induce the unfolding of proteins, allowing the exposure of cleavage sites previously located inside the protein molecules. Thus, hydrolysis of peptides could be enhanced at appropriate temperature [10]. Chotphruethipong, Binlateh, Hutamekalin, Sukketsiri, Aluko, and Benjakul [6] reported that HC generally has a wide range of amino acid composition, sequence, and chain length, which have an impact on functionality and bioactivity. Therefore, mixed proteases (3%P + 4%A) showed an efficient hydrolysis toward salmon skin at pH 8 and 60 °C and the optimal hydrolysis time was 240 min.

#### 3.1.2. Fatty Acid Composition of HC

Fatty acid composition of lipid, which was liberated from skin matrix during hydrolysis prepared under the optimal condition into HC, is shown in Table 1. Lipid in HC constituted at high content (20.87%, dry weight basis). Lipids contained palmitic acid (C16:0) (18.70 g/100 g lipid), oleic acid (C18:1, *n*-9) (16.76 g/100 g lipid), and eicosapentaenoic acid (EPA) (C20:5, *n*-3) (13.07 g/100 g lipid) as the most abundant fatty acids. Lipids extracted from HC consisted of 28.81% saturated (SFA), 30.45% monounsaturated (MUFA), and 40.74% polyunsaturated (PUFA) fatty acids. Unsaturated fatty acid (MUFA and PUFA) content was 71.19% of total fatty acids in HC. EPA was the most abundant PUFA with the content of 13.07%. The result was in line with Aryee, Simpson, Phillip and Cue [11] who found that EPA was the major PUFA in lipids from salmon skin ranging from 12.69 to 14.97%. Additionally, DHA found in lipid from HC was also high (11.10%). This result suggested that lipid from HC was rich in unsaturated fatty acids, which was susceptible to oxidation. Therefore, the fat removal process was required to reduce the fat content in HC in order to lower its oxidation, which mainly causes fishy odor/flavor in resulting HC.

### 3.2. Effect of Centrifugal Cycles on Fat Removal from HC

#### 3.2.1. Fat Content

Fat content of HC from salmon skin hydrolysate after fat removal with DSCS at different cycles (0, 3, 6, and 9 cycles) is presented in Table 2. Generally, fat content in HC without fat removal was 20.87% (dry weight basis). After DSCS was applied, the fat contents of 8.53, 8.44, and 7.80% were obtained for the HC after fat removal for 3, 6, and 9 cycles, respectively. The fat contents of HC subjected to fat removal with DSCS for 3, 6, and 9 cycles were reduced from that of HC without fat removal by 59.14, 59.56, and 62.63%, respectively. Among all HC samples, the HC defatted using DSCS for 9 cycles showed the lowest fat content (*p* < 0.05). Basically, the DSCS has very high G-force and is efficient for separating two liquids. The physical separation takes place within the disc stack, in which the light liquid phase (fat phase) positioning near the bowl axis and the heavy phase (hydrolysate phase) locating at the bowl wall are attained [32]. Therefore, the fat removal with DSCS for 9 cycles lowered fat content in HC from salmon skin without the use of solvents.

#### 3.2.2. Color

The color of powder of HC after fat removal with DSCS for various cycles is presented in Table 2. HC without fat removal obviously had a dark brown color, with the lowest *L**-value and highest *a**- and *b**-values (*p* < 0.05). As the cycles of fat removal by DSCS was increased, the increases in *L**-value along with the decreases in *a**- and *b**-values were attained for HC powders. The highest *L**- and Δ*E**-values, along with lowest *b**-value, were obtained for HC with fat removal for 9 cycles (*p* < 0.05). Color found in HC was related to the amount of pigment cells (chromatophores) in the epidermal layer of salmon skin [33]. Astaxanthin, which is a common fat soluble compound found in salmon, was a main pigment that contributed to the red and orange color in salmon skin [34]. Apart from astaxanthin, melanin was also present in the skin, causing the dark color of HC [33]. In this regard, fat removal process using DSCS could facilitate the removal of pigments in HC solution, especially astaxanthin dissolved in fat, resulting in the increased lightness of resulting HC powder. For melanin, which is water soluble, it might be co-separated with astaxanthin. Melanin could be associated with astaxanthin via hydroxyl group in its structure. This was evidenced by dark color of fat rich fraction after DSCS separation. Additionally, the high fat content could also enhance the discoloration of HC caused by Maillard reaction [16].

### 3.3. Effect of Isopropanol on Fat Removal of HC Powder

HC powder defatted with DSCS for 9 cycles still had high fat content. Therefore, fat extraction with isopropanol of HC powder was further conducted.

#### 3.3.1. Fat Content

Defatting using isopropanol for different cycles (0, 1, 2, and 3 cycles) was applied to the selected HC powder. The efficacy of isopropanol in defatting HC powder was increased with augmenting defatting cycles (Table 3). The higher fat content was observed in the HC powder without defatting with isopropanol (7.78%, dry weight basis), compared to those of HC powders defatted with isopropanol, regardless of cycle used. After defatting with isopropanol for 1 cycle, 1.23% fat (dry weight basis) was found, in which fat was removed by 84.19%, compared to that of HC powder without defatting using isopropanol. Additionally, HC powders defatted with isopropanol for 2 and 3 cycles had no fat detected. Thiansilakul, Benjakul, and Shahidi [31] found that isopropanol was able to remove fat in round scad muscle. Since lipids dispersed in HC were more likely polar in nature, the solvent with slight polarity should be appropriate for the removal of aforementioned lipids.

#### 3.3.2. Color

The color of HC powders defatted with isopropanol for different cycles is shown in Table 3. Generally, the increases in *L**- and Δ*E**-values, along with decreases in *a** and *b**-values, were noticeable for HC powders as the number of defatting cycles was augmented (*p* < 0.05). HC powder defatted with isopropanol for 3 cycles had the highest *L**-value together with lowest *a**-value and *b**-value (*p* < 0.05). Via isopropanol defatting process, fat in the HC powder was more likely removed to a high extent. Moreover, the defatting process using isopropanol might facilitate the removal of pigments, leading to the augmented lightness. Sae-leaw, Benjakul, and O’Brien [16] documented that HC from fish skin had chromatophores rendering color. Additionally, the oxidation of lipid during enzymatic hydrolysis might occur. Lipid oxidation products, especially aldehyde compounds [35], were related with yellow discoloration via the Maillard reaction [16]. Therefore, HC powders defatted with isopropanol for 2 and 3 cycles, which had negligible fat content, were less susceptible to discoloration, as witnessed by the whiter color of HC powder.

#### 3.3.3. Fishy Odor Intensity (FOI)

FOI of HC powder defatted with isopropanol for various cycles is shown in Table 3. All HC powders defatted with isopropanol showed the profound decrease in FOI, compared to HC powder without defatting using isopropanol, regardless of number of cycles used. The lowest FOI was found for HC samples defatted with isopropanol for 2 and 3 cycles (*p* < 0.05). This was in accordance with the marked decrease in fat content and secondary lipid oxidation products (SOPs). It was in agreement with Sae-leaw and Benjakul [36], who reported that the SOPs are one of the major contributors to undesirable fishy odor. Therefore, HC powder defatted with isopropanol for 2 cycles had significantly decreased FOI. 

#### 3.3.4. Peroxide Value (PV)

PV of HC powders without and with defatting by isopropanol for different cycles is shown in Table 3. The highest PV was found for HC powder without defatting (*p* < 0.05). The reduction in PV was generally observed as the number of defatting cycles was augmented (*p* < 0.05). HC powder defatted with isopropanol for 2 and 3 cycles showed the lowest PV (*p* < 0.05). The defatting with isopropanol might have removed fat as well as some primary lipid oxidation products such as hydroperoxides from HC powder more effectively. PV has been used to measure primary lipid oxidation product after H-atom is abstracted and the radicals are combined with oxygen molecules [35]. However, hydroperoxide is not stable and readily decomposed to SOPs. This coincided with low PV found in all samples.

#### 3.3.5. Thiobarbituric Acid Reactive Substances (TBARS) Value

TBARS values of HC powders without and with defatting using isopropanol for various cycles are presented in Table 3. In general, TBARS value of HC powder without defatting using isopropanol was 7.30 mg MDA equivalent/100 g dry HC. Decomposition of hydroperoxides into SOPs might occur in HC during the DSCS fat removal process for 9 cycles, in which air or oxygen could be incorporated into HC containing lipids. However, the defatting with isopropanol for 1–3 cycles could reduce TBARS to 5.22–5.85 mg MDA equivalent/100 g dry HC. Among all the samples, the HC powder defatted with isopropanol for 2 and 3 cycles possessed the lowest TBARS value (*p* < 0.05). This result indicated that the defatting HC powder with isopropanol for at least 2 cycles could remove the SOPs in HC powder more effectively. This was in line with Sae-leaw, Benjakul, and O’Brien [16], who documented that the decreasing TBARS value was obtained in seabass skin after defatting with 30% isopropanol. Nevertheless, some TBARS were still retained in all HC powders after isopropanol defatting process. This might be associated with the aggregation of proteins or peptides during defatting process, which could entrap SOPs. This result was in line with Thiansilakul, Benjakul, and Shahidi [31], who documented that the aggregation of protein was obtained for round scad muscle after defatting with ethanol and isopropanol. As a result, SOPs could not be eliminated from defatted HC powder easily. Lowered TBARS, especially aldehydes, alcohols, and ketones [35], could contribute to a slightly fishy odor in resulting defatted HC powder. However, such a decrease could also enhance storage stability of defatted HC powder. Isopropanol was used for defatting of HC powder. Alcohol is generally capable of dissolving aldehydes. As a result, those aldehydes, the secondary oxidation products, could be leached out by isopropanol washing. 

### 3.4. Characteristics of Selected Defatted HC

#### 3.4.1. Size Distribution

Molecular weight (MW) of peptides in HC powder defatted with isopropanol for 2 cycles as determined by MALDI-TOF mass spectrometry is displayed in Figure 3. Several peaks were observed in spectrum, indicating the presence of various peptides having various MW in HC. All peptides had MW lower than 3000 Da. The peptide with MW of 1460 Da was dominant, followed by those with MW of 1975 and 1534 Da, respectively. This result indicated that the hydrolysis process using mixed proteases (both papain and Alcalase) under optimum conditions could hydrolyze swollen skin and provide a number of low MW peptides in the HC. Additionally, the MW is a key factor governing the biological and functional properties of hydrolysates [37]. Chotphruethipong, Binlateh, Hutamekalin, Aluko, Tepaamorndech, Zhang, and Benjakul [3] documented that the HC from seabass skin with low MW (<3000 Da) could promote osteoblast cell proliferation and enhanced alkaline phosphatase activity (AP-A) during the first 7 days, while inducing mineralization at day 21. It is worth noting that the HC with low MW peptides could be used as functional ingredients or nutraceuticals [37].

#### 3.4.2. Amino Acid Composition

Amino acid compositions of HC powder defatted with isopropanol for 2 cycles are presented in Table 4. HC powder had glycine as the dominant amino acid (359 residues/1000 residues), followed by proline (93 residues/1000 residues) and alanine (80 residues/1000 residues) and glutamic acid/glutamine (65 residues/1000 residues). Tyrosine (8 residues/1000 residues) and hydroxylysine (5 residues/1000 residues) were also found. HC powder contained no cysteine. Benjakul, Karnjanapratum, and Visessanguan [1] reported glycine (326 residues/1000 residues), alanine (128 residues/1000 residues), and proline (112 residues/1000 residues) in the HC from seabass skin. Glycine generally occurs at every third position in collagen polypeptide and imino acids (proline and hydroxyproline) are also constituted [38]. HC powder consisted of 20.90% essential amino acids. Moreover, the lower content of imino acids (137 residues/1000 residues) in defatted HC from salmon skin was observed than that of HC from seabass skin (192 residues/1000 residues). For hydrophobic amino acids, the defatted HC showed 64.10%, while HC from seabass skin constituted 67.48% hydrophobic amino acid as reported by Chotphruethipong, Aluko, and Benjakul [30]. This was plausibly governed by different species and processes used for HC production. It was postulated that the defatting process with isopropanol could remove free imino acids and hydrophobic amino acids such as proline, hydroxyproline, alanine, and leucine in HC powder to some degree. Thus, defatting using solvent likely affected amino acid composition of HC to some extent.

#### 3.4.3. Proximate Compositions and Mineral Content

Table 5 shows proximate compositions of HC powder defatted with isopropanol for 2 cycles. Typically, HC powder was hygroscopic, related with the charged or polar residues exposed after hydrolysis [39]. The resulting HC powder was rich in protein (94.72%, dry weight basis), indicating that proteins in HC became more concentrated after the fat removal processes [40]. In addition, ash content of HC was 5.08% (dry weight basis). This was possibly associated with salt formed during neutralization between acid and alkaline solutions during hydrolysis [8]. This was confirmed by high sodium content (2.50%, dry weight basis) (Table 5). Moreover, the salmon scale contained high amount of mineral, which could also be released to HC during hydrolysis as indicated by the presence of calcium and phosphorous in HC at 0.32 and 0.31% (dry weight basis), respectively. Ca-hydroxyapatite has been known to be the major mineral complex in fish scales [41].

#### 3.4.4. Volatile Compounds

Volatile compounds in HC powder defatted with isopropanol for 2 cycles (HC-C9/ISP2) in comparison with the HC defatted using centrifugal separator for 9 cycles (HC-C9) and HC without fat removal are shown in Table 6. Aldehydes are the most prevalent volatile SOPs in HC and have been used as the index of lipid oxidation in foods due to their low odor threshold value. They majorly contribute to off-odor and off-flavor [42]. Ross and Smith [42] documented that several aldehydes generated during oxidation were octanal, nonanal, pentanal, hexanal, etc. Among all aldehydic compounds in the present study, nonanal was the most dominant in HC, followed by undecanal, 2,6-nonadienal, octanal, and heptanal, respectively (Table 6). Principally, *n*-alkanals and 2-alkenals and 2,4-alkadienals occurred from oxidation of *n*-6 or *n*-9 unsaturated fatty acids, especially oleic acid [43]. Salmon skin contained oleic acid at a high level, 16.76 g/100 g lipid (Table 1). This led to a large number of aldehydes formed in HC. HC-C9 generally had a higher abundance of aldehydes than HC. This might be associated with higher lipid oxidation occurring during fat removal with DSCS for 9 cycles. The long exposure to centrifugal shearing force of DSCS might decrease oil droplet size with coincidentally increasing surface area of retained fat in HC solution. As a consequence, lipid oxidation could be enhanced. Nevertheless, the abundance of all aldehydes was lowered in HC-C9/ISP2 in comparison with the HC and HC-C9. No 2-pentenal, 4-heptenal, 2-methyl-2-heptenal, 2-octenal, 2-propyl-2-heptenal, and 2,4-heptadienal were detected in HC-C9/ISP2. This result suggested that the defatting of HC powder with isopropanol for 2 cycles could lower the amount of aldehydes. This was related with lower PV and TBARS, as documented in Table 3. Regarding the FOI, Varlet et al. [44] documented that carbonyl compounds, especially aldehydes involving 4-heptenal, octanal, decanal, and 2,4-decadienal, mainly contributed to fishy odor in salmon flesh. The lower abundance of octanal, along with disappearance of 4-heptenal, was observed for HC-C9/ISP2, compared to those of HC and HC-C9/ISP2 samples. It is worth noting that the lower FOI was obtained for HC powder defatted with isopropanol for 2 cycles (Table 3). This was in line with Chang, et al. [45], who documented that the use of 35–75% isopropanol was very efficient in reducing volatile compounds, particularly aldehydes in lentil protein isolate.

Alcohols including 1-dodecen-3-ol, 1,5-octadien-3-ol, and 7-hexadecyn-1-ol were found in HC. HC-C9 contained 2,4-octadien-1-ol, 1-octen-3-ol, 1,5-octadien-3-ol, 1-nonanol, and 2-ethylnon-1-en-3-ol. Alcohols are known as the SOPs and contribute to off-flavor because of their low odor threshold [42]. However, the alcohol was not detected in HC-C9/ISP2 sample (Table 6). Furthermore, ketones, another SOPs [28], including 3-octanone, and 3,5-octadien-2-one were noted in HC and HC-C9, but were not present in HC-C9/ISP2. Lower levels of 2,3-octanedione and 3-undecen-2-one were found in HC-C9/ISP2, compared to HC-C9. This confirmed the lowered ketones in HC-C9 by isopropanol extraction.

Overall, the lower SOPs involving aldehydes, alcohols, and ketones coincided with the decreased PV and TBARS values (Table 3). The results suggested that SOPs were lower in HC-C9/ISP2, compared to HC and HC-C9. Thus, the fat removal via DSCS process for 9 cycles, followed by treatment using isopropanol for 2 cycles, was effective in the removal of fat, which was a precursor for oxidation as well as in eliminating SOPs associated with offensive fishy odor in resulting HC.

## 4. Conclusions

The production of fat-free hydrolyzed collagen (HC) powder from salmon skin with negligible fishy odor was achieved. One-step hydrolysis using mixed proteases (3% papain and 4% Alcalase) at pH 8 and 60 °C for 240 min was used for HC production. HC powder from hydrolysate with fat removal using DSCS for 9 cycles had the lower fat content with higher lightness, compared to that without fat removal. HC powder was subsequently defatted with isopropanol for 2 cycles (HC-C9/ISP2). The obtained HC powder was free of fat with lowered fishy odor and lipid oxidation products but showed the highest *L**-value (84.52) (*p* < 0.05). HC-C9/ISP2 contained peptides with MW less than 3000 Da. HC-C-/ISP/2 had high protein content with high amounts of glycine and imino acids. HC-C9/ISP2 had the lower abundance of odorous compounds, compared to that without the fat removal process. Therefore, the developed HC powder lacking in fat and fishy odor could be potentially applied in food or pharmaceutical products.

## Figures and Tables

**Figure 1 foods-10-02257-f001:**
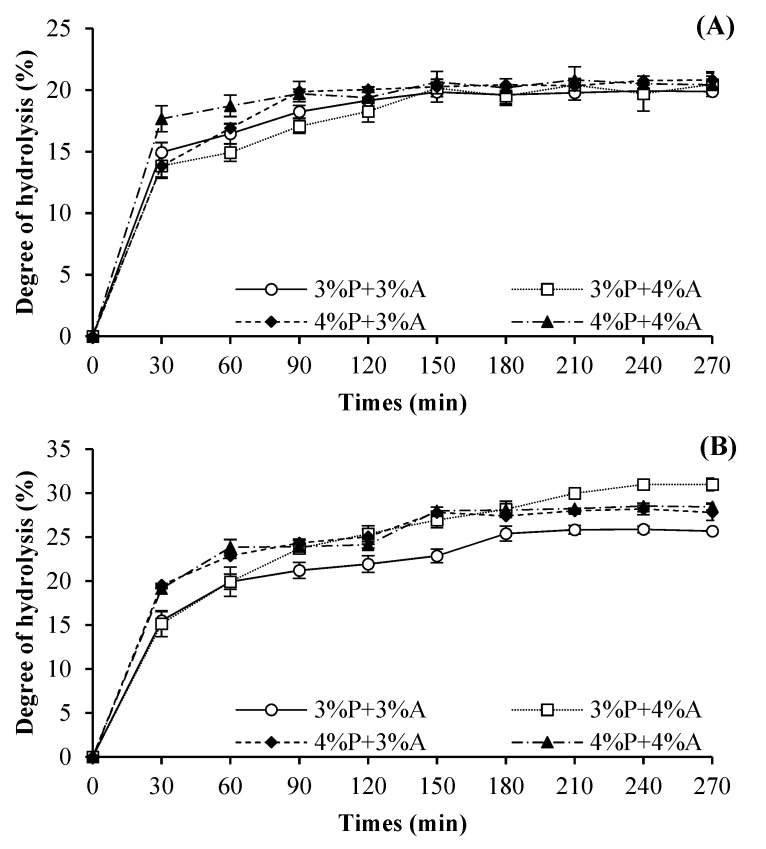
Degree of hydrolysis (DH) curve of hydrolyzed collagen (HC) from salmon skin prepared using mixed proteases at different concentrations at pH 7.0 (**A**) and 8.0 (**B**). P: papain; A: Alcalase. Bars represent the standard deviation (*n* = 3).

**Figure 2 foods-10-02257-f002:**
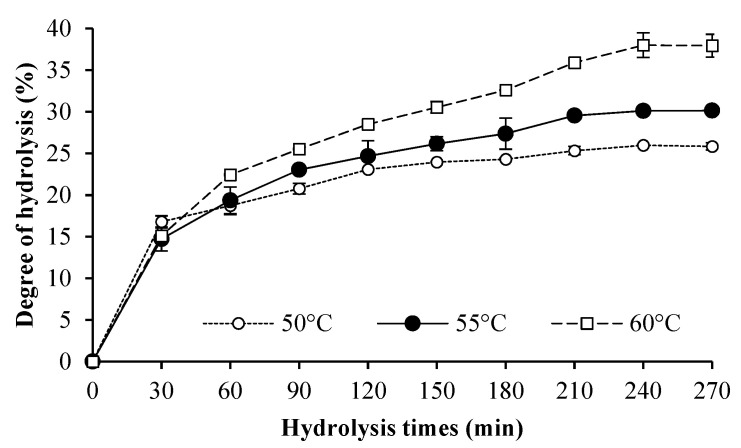
DH curve of HC from salmon skin prepared at different temperatures. Hydrolysis was conducted at pH 8 using mixed proteases (3%P + 4%A) for 270 min. Bars represent the standard deviation (*n* = 3).

**Figure 3 foods-10-02257-f003:**
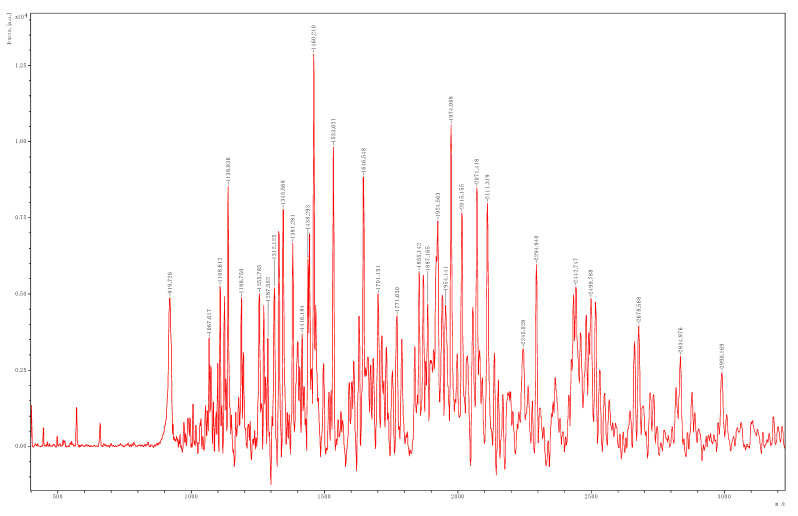
Size distribution of HC powder defatted using DSCS for 9 cycles, followed by washing using isopropanol for 2 cycles determined by MALDI-TOF mass spectrometry.

**Table 1 foods-10-02257-t001:** Fatty acid composition of lipid extracted from hydrolyzed collagen without fat removal.

Fatty Acids	Content (g/100 g Lipid)
C14:0	4.30 ± 0.06
C15:0	0.49 ± 0.02
C16:0	18.70 ± 0.07
C16:1	7.83 ± 0.20
C18:0	4.34 ± 0.01
C18:1 *n*-9	16.76 ± 0.05
C18:2 *n*-6	11.69 ± 0.02
C20:1 *n*-9	2.45 ± 0.01
C18:3 *n*-3 (ALA)	0.71 ± 0.01
C20:2 *n*-6	1.97 ± 0.01
C22:1 *n*-9	1.94 ± 0.01
C23:0	0.98 ± 0.00
C22:2 *n*-6	2.20 ± 0.02
C20:5 *n*-3 (EPA)	13.07 ± 0.03
C24:1 *n*-9	1.46 ± 0.02
C22:6 *n*-3 (DHA)	11.10 ± 0.05
Saturated fatty acids (SFA)	28.81 ± 0.11
Monounsaturated fatty acid (MUFA)	30.45 ± 0.16
Polyunsaturated fatty acid (PUFA)	40.74 ± 0.06

Values are presented as mean ± SD (*n* = 3).

**Table 2 foods-10-02257-t002:** Fat content and color of hydrolyzed collagen without and with fat removal using disk stack centrifugal separator (DSCS) for different centrifugal cycles.

Parameters	Cycles			
	0	3	6	9
Fat content (%, dry weight basis)	20.87 ± 0.45 a	8.53 ± 0.17 b	8.44 ± 0.08 b	7.80 ± 0.14 c
*L**	55.07 ± 0.44 d	71.81 ± 0.30 c	76.73 ± 0.49 b	79.05 ± 0.31 a
*a**	3.88 ± 0.10 a	3.17 ± 0.08 b	1.54 ± 0.02 c	1.44 ± 0.03 c
*b**	25.36 ± 0.38 a	20.99 ± 0.66 b	16.30 ± 0.07 c	15.50 ± 0.08 d
∆*Ε**	-	17.33 ± 0.15 c	23.60 ± 0.43 b	26.04 ± 0.25 a

Values are presented as mean ± SD (*n* = 3). Different lowercase letters in the same row indicated significant differences (*p* < 0.05).

**Table 3 foods-10-02257-t003:** Fat content, color, fishy odor intensity, peroxide value (PV), and thiobarbituric acid reactive substances (TBARS) of hydrolyzed collagen powder defatted with isopropanol for different cycles.

Parameters	Cycles			
	0	1	2	3
Fat content (%, dry weight basis)	7.78 ± 0.19 a	1.23 ± 0.13 b	ND	ND
*L**	78.98 ± 0.17 d	80.06 ± 0.60 c	84.52 ± 0.35 b	86.51 ± 0.33 a
*a**	1.44 ± 0.02 a	1.38 ± 0.08 ab	1.29 ± 0.08 b	1.17 ± 0.09 c
*b**	15.47 ± 0.04 a	14.78 ± 0.46 b	13.37 ± 0.11 c	12.11 ± 0.56 d
Δ*E**	-	1.41 ± 0.38 c	5.93 ± 0.31 b	8.26 ± 0.45 a
Fishy odor intensity	2.37 ± 0.21 a	1.82 ± 0.24 b	1.49 ± 0.24 c	1.41 ± 0.29 c
PV (mg cumene hydroperoxide equivalents/100 g dry HC)	4.38 ± 0.10 a	2.71 ± 0.01 b	1.05 ± 0.04 c	1.03 ± 0.04 c
TBARS (mg MDA equivalents/100 g dry HC)	7.30 ± 0.20 a	5.85 ± 0.07 b	5.40 ± 0.06 c	5.22 ± 0.09 c

Values are presented as mean ± SD (*n* = 3). Different lowercase letters in the same row indicated significant differences (*p* < 0.05). ND: not detected.

**Table 4 foods-10-02257-t004:** Amino acid composition of hydrolyzed collagen powder defatted using DSCS for 9 cycles, followed by using isopropanol for 2 cycles.

Amino Acids	Content (Residues/1000 Residues)
Alanine	80
Arginine	52
Aspartic acid/asparagine	55
Glutamic acid/Glutamine	65
Glycine	359
Histidine	14
Hydroxylysine	5
Hydroxyproline	44
Isoleucine	14
Leucine	32
Lysine	43
Methionine	15
Phenylalanine	14
Proline	93
Serine	50
Threonine	32
Tryptophan	1
Tyrosine	8
Valine	24
Total amino acids	1000
Imino acids (Hyp + Pro)	137

**Table 5 foods-10-02257-t005:** Proximate compositions of hydrolyzed collagen powder defatted using DSCS for 9 cycles, followed by washing using isopropanol for 2 cycles (HC-C9/ISP2).

Proximate Compositions	Content (%, Dry Weight Basis)
Protein	94.72 ± 0.06
Ash	5.08 ± 0.07
Sodium (Na)	2.50 ± 0.05
Calcium (Ca)	0.32 ± 0.01
Phosphorous (P)	0.31 ± 0.00

Values are presented as mean ± SD (*n* = 3).

**Table 6 foods-10-02257-t006:** Volatile compounds of hydrolyzed collagen without and with different fat removal processes.

Compounds	HC	HC-C9	HC-C9/ISP2
**Aldehyde**			
Hexanal	0.19	0.19	0.08
2-Pentenal	0.07	ND	ND
Heptanal	0.50	0.35	0.18
4-Heptenal	0.11	0.02	ND
Octanal	0.64	1.09	0.47
2-Methyl-2-heptenal	0.11	0.10	ND
Nonanal	5.11	7.42	4.53
2-Octenal	0.18	0.29	ND
2-Propyl-2-heptenal	0.16	0.10	ND
2,4-Heptadienal	0.43	0.35	ND
2-Nonenal	0.27	0.37	0.35
2,6-Nonadienal	0.83	0.82	0.70
Undecanal	3.65	5.02	2.38
2-Decenal	0.17	0.54	0.44
**Alcohol**			
2,4-Octadien-1-ol	ND	0.01	ND
1-Octen-3-ol	ND	0.19	ND
1-Dodecen-3-ol	0.24	ND	ND
1,5-Octadien-3-ol	0.09	0.09	ND
1-Nonanol	ND	0.14	ND
2-Ethylnon-1-en-3-ol	ND	0.64	ND
7-Hexadecyn-1-ol	0.27	ND	ND
**Ketone**			
3-Octanone	0.21	0.11	ND
2-Methyl-3-octanone	0.53	ND	ND
2,3-Octanedione	ND	0.56	0.28
3,5-Octadien-2-one	0.82	0.78	ND
3-Undecen-2-one	0.78	2.59	1.22

Values are expressed as abundance (×10^7^). ND: not detectable. HC-W/O: HC without fat removal processes; HC-C9: HC with fat removal using DSCS for 9 cycles; HC-C9/ISP2: HC defatted with DSCS for 9 cycles, followed by washing using isopropanol for 2 cycles.

## Data Availability

Data are not shared.

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
