# Peer review of "Development of Hydrolysis and Defatting Processes for Production of Lowered Fishy Odor Hydrolyzed Collagen from Fatty Skin of Sockeye Salmon (Oncorhynchus nerka)"

_foods, 2021, doi:10.3390/foods10102257_

Round 1
Reviewer 1 Report
This manuscript is an important contribution to the development of Fat-free hydrolyzed collagen powder with lowering fishy odor that could be potentially applied in food or pharmaceuticals products. However, this manuscript is necessary to be improved following the comments below.
Abstract:
The abstract is poorly written. The abstract should contain an introductory sentence, brief method (one or two sentences), brief result and a summary sentence at the end.
Introduction:
The introduction should be improved with updated reports from the literature.
Discussion:
There is no discussion in this manuscript. The authors must include a discussion section.
Author Response
Thank you very much for your invaluable comments and suggestion. All queries have been responded and the corrections have been provided in text as highlighted in yellow.

Reviewer 2 Report
This is an interesting paper focusing on a rest raw material and its stability.
Please find my comments and suggestions starting with the beginning of the manuscript.
L2-3. Title: Should the title reflect the salmon species that are studied?
L15-17. This is an unclear sentence with an un-described abbreviation and where "Lowest fishy odour..." should be explained better (compared to what).
L20. The sentence “Na, Ca and P were found” seems a bit misplaced in the abstract
L32. Should “% papain” show if this is in weight% or not?
L36-44 The statements in this part should be better referred such as why the “…salmon is the most popular fish” and references on the taste and smell challenges are lacking. L40 - only one reference on the enzymatic hydrolysis is reported. There is more relevant research on this area such as for the Atlantic salmon.
L36. I suggest that "popular fish" are replaced by "popular fish species"
L39. "Particularly skins" should be replaced by "such as" since there are also other major rest raw materials from salmon.
L42. "Peptides with high bioactivity" shows one reference and this one is on antioxidant activity. This sentence should therefore reflect this limitation or add more references that can support the statements. That the scientific evidence of biological antioxidative effect is limited should be reflected in the text.
L46.Should “implemented” be changed to “tested” since the process is not being used?
L75-76. The procedure should be described better for others to repeat. Unclear sentences - It is unclear if the muscle from skin were washed at <10C or the part without meat? How was the packaging? Vacuum or with air? This is important when working with resources that are susceptible to lipid oxidation.
L72-86. High temperatures are used in all processes leading to increased lipid oxidation during the handling procedure. Can the author explain why these high temperatures are used?
L109, L122 and L141. Two different methods for extracting lipids are used. The Bligh & Dyer are known to extract all lipid classes and the AOAC method might discriminate the polar lipids. On L141 it refers to methods above. This should be clear what methods are used.
L201 and L203. Be consequent in the use of words “were applied respectively” and were used respectively”.
General it is a mix between what others have found and what this work has found that should be clarified in the manuscript. This is both in the materials and methods (specially page 5) and in section 3.1-3.4.
L266 -268. This is an unclear sentence. It is unclear what is meant by “was lowered than that of HC by 59,14, 59, 56 and 62,63% respectively”. This should therefore be written to clarify what it means.
L281. Parts of the text need some language improvement specially on the use of past and present such as “Astaxanthin which was” to “Astaxanthin which is”
L303. Explain better “The higher fat content….” higher compared to what?
L337-338. Unclear sentence “….minimized FOI”
L344. Replace “least PV” with “lowest PV”
L350-367 (L443). Should this part discuss better the washing procedure with isopropanol effect on the removal of aldehydes and thereby effect the stability? Are these oxidation products washed out since they are soluble in the solvent?
L363-366. This part should be better explained and maybe linked to what has been found in the referred paper.
L242 Table 5 uses the abbreviation HC-C9/ISP2 without explaining what it means.
L432-433. This part needs to be improved to clarify what others have done and what has been done in this work. Using words such as: In their work they found that….etc. and In the present work…... will help the reader to differentiate between this new findings and other research.
Minimum 19 of the 43 references are from the same author. Specially the general part/introduction should add some more variety in the reference basis.
Author Response
Thank you for your understanding and support. The suggestions and comments have been responded and the corrections have been included in text as highlighted in green.

Round 2
Reviewer 1 Report
N/A